# OpenReview forum: "CUPID: Evaluating Personalized and Contextualized Alignment of LLMs from Interactions"
_colmweb.org/COLM/2025/Conference — COLM 2025_

### Official Review · Reviewer_CQeC · 2025-05-10

**Rating:** 6
**Confidence:** 4
**Ethics Flag:** 1

**Summary:**

This paper proposes LILAC, a benchmark for evaluating the capabilities of LLMs which infer the user’s contextual preference and generate a response satisfying the user’s preference. For this purpose, this paper creates the dialogue data concern user’s persona, contextual factor, and contextual preferences which contain consistent, contrastive, and changing instances. Moreover, this paper evaluates the capabilities of 10 LLMs’ inferring and generating from Preference Match and Preference Alignment metrics. For the results of evaluation, this paper gives some significant conclusions.

**Questions To Authors:**

1. Figure 2 is not mentioned in the paper. Did I not see it or did you forget?
2. According Line 91 “Si = (ci, pi, Di)”, Line 94 “We assume that each session’s context is only defined and influenced by a single factor”, and Line 173 “each factor-preference pair”, does it mean that ci and pi are one-to-one corresponding in each Si?
3. You utilized zero-shot with prompts for LLMs to infer users’ references and generate responses. What would be the effect if you fine-tuned the LLMs using your dataset? Have you considered prompt fine-tuning?

**Reasons To Accept:**

1. This paper’s work of evaluating LLMs from the capabilities of inferring user’s contextual preference and generating response for the preference is interesting.
2. The LILAC benchmark created by this paper is helpful for further research work of evaluating a model’s capabilities of inferring user’s preference and generating response, and the Data Generation Pipeline is characteristic.
3. This paper leverages the text and appendix to give a wealth of information, especially dialogue data instances generation, benchmark creation, and experiment setup.

**Reasons To Reject:**

1. The evaluation of the inferring and generation capabilities of LLMs is not a new research issue in itself, although the evaluation for the capability of inferring user context preferences and generating responses is distinctive.
2. Although the evaluation itself is meaningful, the purpose of the evaluation is to provide some suggestions from the perspective of LLMs application, bur this paper lacks effective suggestions.
3. For the LILAC benchmark created by this paper, it is not enough if only used to evaluate LLMs. Moreover, it is not enough only using one dataset to evaluate LLMs, too.
4. In Section 5, Personalized LLMs and Interactions with LLMs are related works with this paper’ work, however, more related work is the evaluation for LLMs, this paper lacks the discussion of this aspect of works.
5. What mean the Desiderata 1, Desiderata 2, and Desiderata 3? The Desiderata should be given definition or statement clearly. The current form of expression is puzzling.
6. A period “.” is missing at the end of Line 56. Please check carefully for similar problems.

---

> ### Author Response · Authors · 2025-06-03
>
> Thank you very much for the thorough comments and feedback!
>
> > **evaluation of the inferring and generation capabilities of LLMs is not a new research issue in itself**
>
> We agree that evaluating LLMs’ general inference and generation capabilities is not novel by itself. Instead, as also pointed out by this review, our work’s novelty lies in specifically evaluating LLMs’ capabilities at (1) inferring user contextual preferences from prior interactions, and (2) generating responses that can satisfy these contextual preferences.
>
> > **purpose of the evaluation is to provide some suggestions from the perspective of LLM applications**
>
> If the paper gets accepted, with the extra page for the camera-ready, we will add a discussion subsection with suggestions for LLM applications based on our findings: (1) retrieve contextually relevant prior interaction sessions when responding to new user requests; (2) storing and retrieving summaries of prior sessions when using smaller LLMs; and (3) further prompt or model developments to better infer user preferences from multi-turn interactions.
>
> > **it is not enough if only used to evaluate LLMs… not enough only using one dataset to evaluate LLMs**
>
> This is a great point. We agree that datasets are of greater utility when they can be used for both training and testing. While LILAC is intended solely for evaluation, we will release the data generation pipeline, which can be used to easily create separate training sets to fine-tune LLMs.
>
> We agree that multiple datasets should be used to evaluate LLMs. Our dataset, which focuses on assessing a specific capability (i.e., inferring and applying users’ contextual preferences), should be used alongside other benchmarks to more comprehensively assess models’ performance.
>
> > **more related work is the evaluation for LLMs**
>
> Based on this suggestion, we plan to add a Related Work subsection on LLM evaluation methods and benchmarks. While the current subsections cover evaluation related to Personalized LLMs and Interaction with LLMs, a separate section on general LLM evaluation methods can help position our work.
> For example, evaluation on user preferences [1], LLM-as-a-Judge [2], evaluation through user simulations [3], evaluation on synthetic datasets [4], etc.
>
> > **The Desiderata should be given definition or statement clearly**
>
> Section 3 begins by defining three Desiderata (i.e., requirements) for our dataset and then refers to them through the section to note how each design choice satisfies these requirements (e.g., "Desiderata 1"). We agree that this presentation can be unclear and will explicitly explain this when first referring to a Desiderata in the section.
>
> > **A period “.” is missing … / Figure 2 is not mentioned in the paper**
>
> Thank you for pointing out these issues. We will ensure to revise and double-check the camera-ready version.
>
> > **does it mean that ci and pi are one-to-one corresponding in each Si**
>
> Yes. There is a one-to-one correspondence between the context factor $c_i$ and preference $p_i$ in each interaction session $S_i$. Multiple sessions can share the same context factor and relevant user preference.
>
> > **effect if you fine-tuned the LLMS using your dataset**
>
> Similar to prior work [5, 6], we focus on evaluating the “out-of-the-box” capabilities of LLMs at the proposed tasks. As a result, we did not experiment with fine-tuning or prompt tuning on the dataset. While we expect that tuning on the dataset (or additional samples generated from our pipeline) would improve performance, we leave this for future work.
>
> [1] https://arxiv.org/abs/2403.04132
>
> [2] https://arxiv.org/abs/2306.05685
>
> [3] https://arxiv.org/abs/2310.11667
>
> [4] https://aclanthology.org/2023.emnlp-main.890/
>
> [5] https://arxiv.org/abs/2410.10813
>
> [6] https://openreview.net/pdf?id=YfHxQSoaWU

---

> > ### Comment · Reviewer_CQeC · 2025-06-06
> >
> > Thank you for your serious and detailed rebuttal.
> > I admit your contribution and at the same time clarify some issues.
> > I believe you will revise the paper carefully, but the current version is flawed.

---

> > > ### Author Response · Authors · 2025-06-08
> > >
> > > Thank you for your response! We are strongly committed to revising the paper according to your suggestions. We sincerely appreciate your time and dedication in reviewing our paper.

---

> > > > ### Comment · Reviewer_CQeC · 2025-06-10
> > > >
> > > > Thank you again for your responses.

---

### Official Review · Reviewer_sNUJ · 2025-05-12

**Rating:** 8
**Confidence:** 3
**Ethics Flag:** 1

**Summary:**

This paper addresses the challenge of aligning large language models (LLMs) with human preferences, emphasizing that such preferences are often context-dependent rather than static. To tackle this, the authors introduce LILAC (Learning from Interactions for LLM Alignment in Contexts), a benchmark designed to evaluate LLMs’ ability to infer user preferences from prior interaction histories. The benchmark includes 756  interaction histories, each consisting of multiple dialogue sessions between a simulated user and an LLM. Given a new request and the preceding context, models are assessed on their ability to (1) infer context-sensitive user preferences and (2) generate satisfactory responses. Evaluation across 10 open and proprietary LLMs shows that current models struggle with this task, with none exceeding 50% precision or 65% recall.

**Questions To Authors:**

Will data be made available to the community  ?

**Reasons To Accept:**

- Introduces an original and well-designed benchmark for context-aware preference detection and in-context response generation

- Clearly defined tasks and well-formalized evaluation metrics

**Reasons To Reject:**

The benchmark relies entirely on a synthetic data generation pipeline. While the authors do provide a data validation analysis (Section 3.2 and Appendix E), and include example interactions, some examples feel somewhat abstract or theoretical. It would strengthen the paper to further discuss the practical relevance of the benchmark—specifically, how common such context-aware preference alignment scenarios are in real-world applications, and in which application use cases this benchmark is most applicable.

---

> ### Author Response · Authors · 2025-06-03
>
> Thank you for the positive review and the constructive feedback!
>
> > **further discuss the practical relevance of the benchmark**
>
> Our benchmark reflects long-term and frequent use of LLM-based chat assistants, like ChatGPT. Through frequent interactions, the user may provide the assistant with diverse requests that may share contexts. For example, a user can ask the following requests to an assistant: write an email to PhD advisor requesting time off, write another one asking for paper feedback, revise an email responding to the feedback, etc. If the assistant recognizes that these requests share the same context (PhD advisor), it can infer the user’s preference for this context from prior interactions and personalize future responses.
>
> > **available to the community?**
>
> Yes. We are excited to publicly release our dataset. Additionally, we will release the code for the data generation pipeline and the evaluation suite (including the fine-tuned metric).

---

> > ### Comment · Reviewer_sNUJ · 2025-06-04
> > **Acknowledgement of authors' response**
> >
> > This is my acknowledgement of authors' response. Good to know the dataset will be released. No modifications of my score which already reflects well my feedback on the paper.

---

> > ### Author Response · Authors · 2025-06-08
> >
> > Thank you for your response and the positive endorsement! We sincerely appreciate your time and dedication in reviewing our work.

---

### Official Review · Reviewer_puYD · 2025-05-13

**Rating:** 8
**Confidence:** 4
**Ethics Flag:** 1

**Summary:**

This paper proposes a benchmark that evaluates the capabilities of ten leading open/proprietary LLMs in inferring and satisfying contextual and dynamic user preferences. The benchmark consists of 756 LLM simulated and human verified conversational interactions, and associated request, context factor, contextual preference as well as prior interactions are provided for each of the conversational interactions.

**Questions To Authors:**

- It would be great if the authors could discuss in their paper regarding what are possible pathways and potential challenges in building LILAC-like benchmark from resources that have real user interactions and feedback like [1].
- Figure 2 may need to be recomplied (probably compressed a little bit) as it causes rendering issue on Mac devices when using Preview to read the pdf.

[1] https://arxiv.org/pdf/2408.15549

**Reasons To Accept:**

- The benchmark focuses on contextual and dynamic preferences, which is a novel aspect in terms of conversational personalization and preference alignment.
- Results and error analysis around consistent, contrastive, and changing preferences provide many new insights that are worth sharing with the community.

**Reasons To Reject:**

- The benchmark heavily relies on synthetic data generated by one model (Claude 3.5). The consequences include but not limited to bias towards Claude models and overlook that real users often express preferences more implicitly and subtly.
- Although the dataset is mostly LLM‑generated, its size is very small. It is unclear if the synthetic data can be used directly for model alignment or if it's possible to automate the verification process.

---

> ### Author Response · Authors · 2025-06-03
>
> Thank you for the positive and insightful comments!
>
> > **bias towards Claude models**
>
> This is an important point. To assess the presence and significance of potential self-enhancement bias, we conducted an analysis (Appendix F.5) where we used our pipeline to synthesize small datasets with three other models: Claude 3.7 Sonnet, GPT-4o, and DeepSeek-R1. Each of these models was evaluated on each other’s datasets to test whether a model performed better on data that it generated. Results showed no clear evidence of self-enhancement: Claude 3.7 Sonnet outperformed on the Inference task in all of the other models’ datasets, and DeepSeek-R1 outperformed in the Generation task in all datasets. This analysis of self-enhancement bias was based on similar analyses from prior work [1, 2].
>
> > **overlook that real users often express preferences more implicitly and subtly**
>
> We agree with this point. Our benchmark was designed with more explicit signals from simulated users to assess whether LLMs struggle even in these “easier” cases. Our results show that they do struggle and reinforce the need for further model developments to adequately provide personalized and contextual support in real-world applications.
>
> > **the dataset … size is very small … / unclear if … can be used directly for model alignment / possible to automate the verification process**
>
> Dataset size was limited by the cost of human verification. We opted for a relatively smaller but more rigorously and carefully validated dataset. This thorough verification revealed that ~9% of generated samples required manual correction. Given this relatively low error rate, we plan to open-source the data generation pipeline to support large-scale data generation for model training and alignment. Also, as most errors were related to the current request revealing the user’s preference, the pipeline can be extended with a simple LLM-based validation module that can identify and revise such cases—as suggested by Reviewer puYD. We believe that this can help further reduce errors and increase the overall data quality for alignment.
>
> > **possible pathways and potential challenges in building .. from resources that have real user interactions**
>
> This is a very interesting point. If the paper gets accepted, we will use the extra page for the camera-ready to add a short discussion on how to extend the data generation pipeline to be grounded on real user data (e.g., WildChat/WildFeedback). Specifically, user requests in these datasets can be used to seed the generation of other requests with the same or different contextual factors—creating interaction session histories based on realistic requests but controlling for the instance type (e.g., Consistent, Contrastive). Similarly, the simulated user in our pipeline can be guided with real user messages to produce more realistically styled feedback.
>
> > **Figure 2 may need to be recomplied**
>
> Thank you for pointing this out! We will make sure to revise this.
>
> [1] https://openreview.net/forum?id=3GTtZFiajM
>
> [2] https://arxiv.org/abs/2412.10424

---

> > ### Comment · Area_Chair_5eKb · 2025-06-06
> > **Acknowledging Author Response**
> >
> > Dear Reviewer puYD,
> >
> > Even though you are recommending accepting this paper, please take a moment to acknowledge the author's response and ask any follow-up questions that you may have. Reviews can help authors improve their submissions.
> >
> > Your AC

---

> > ### Comment · Reviewer_puYD · 2025-06-07
> >
> > Sorry for the delay in my response due to my busy workday schedule.
> > The analysis mitigated my concern about self-enhancement bias. It would be great if the authors could also release a larger, unverified version of the dataset alongside the code, which would make it easier for the community to utilize the data. Assuming the authors would make adequate revisions and open-source their work, I have updated my rating to 8.

---

> > ### Author Response · Authors · 2025-06-08
> >
> > Thank you for your response! We are strongly committed to carrying out the revisions. As you suggested, we will also look at releasing a larger, unverified dataset for the community.
> >
> > We sincerely appreciate your time and dedication in reviewing our work.

---

### Comment · Area_Chair_5eKb · 2025-06-06
**Discuss Period Ends on June 10th**

Dear Reviewers,

The discussion period ends on June 10th. If you have any other follow-up questions or concerns, then please post them soon so authors have a chance to respond. Please also look at other reviews and follow-up messages.

Thank you
Your AC

---

### Decision · Program_Chairs · 2025-07-08

**Decision:**

Accept

**Comment:**

This paper introduces a dataset, LILAC to detect context-dependent user preferences from interactions and using them to generate appropriate responses. The dataset has 756 sessions of interaction between users and LLM-assistants with human curations. The main finding is that the proprietary LLMs are not good at inferring user preferences from interaction history.

Reviewers liked this paper, with two reviewers making strong recommendations. The main drawback, as highlighted by reviewer CQeC, is that the paper lacks a suggestion to improve LLM practice. I agree with this conclusion, but also think that the benchmark will still prove useful for studying and detecting user preferences, especially in the personalization literature. I can imagine that others will use this benchmark to design better agents or learning approaches to personalize LLMs. If the paper does get accepted, I encourage authors to add a section on suggestions for improving the inference of user preferences (authors have indicated that they will add suggestions).

Overall, I am suggesting accepting the paper.